# Silver Nanoparticles: Mechanism of Action and Probable Bio-Application

**DOI:** 10.3390/jfb11040084

**Published:** 2020-11-26

**Authors:** Ekaterina O. Mikhailova

**Affiliations:** Institute of innovation management, Kazan National Research Technological University, K. Marx Street 68, 420015 Kazan, Russia; katyushka.glukhova@gmail.com

**Keywords:** silver nanoparticles, AgNPs, green synthesis, bio-application

## Abstract

This review is devoted to the medical application of silver nanoparticles produced as a result of “green” synthesis using various living organisms (bacteria, fungi, plants). The proposed mechanisms of AgNPs synthesis and the action mechanisms on target cells are highlighted.

## 1. Introduction

In the modern world, “green technologies” are gaining more and more popularity due to their effectiveness, non-toxicity, and “eco-friendly”. One of the directions of “green synthesis” is the production of elementary silver nanoparticles (AgNPs) for use in various areas of human activity, primarily in medicine. It should be noted that mankind has been familiar with the bactericidal effects of Ag^+^ ions from time immemorial. The presence of both silver ions and AgNPs was established by the research in the solution named “Holy water”, known from the beginning of the first Millennium as a protection tool against infection by microorganisms [1]. It is thanks to the silver ions and AgNP suspension that it can have a bactericidal, bacteriostatic, antiviral, and antifungal effect on a large number of pathogenic microorganisms, yeast fungi, and viruses. In addition, the inhibitory effect can sometimes be expressed even slightly stronger in comparison with penicillin, biomycin, and other “classic” antibiotics due to the resistance of many strains of microorganisms to antibiotics [2,3]. This circumstance, together with their low toxicity, almost complete absence of allergic reactions, and good tolerance, has made AgNPs a very popular survey object. Moreover, the high interest in silver nanoparticles over the past decades has allowed for not only the confirmation of their antibacterial activity, but also to discover new properties worthy of application, which will be discussed below.

Anticipating the story of the use of AgNPs with various properties used in practice, it is necessary to note the wide and diverse forms of nanoparticles obtained during their synthesis by both physico-chemical and biological methods. The variety of objects used for the synthesis of nanoparticles inevitably leads to a variety of AgNPs forms: these can be “nanowires”, tabular prisms, cubes, octahedra, and pyramids [4,5]. The different studies devoted to silver nanoparticles showed that the shape and size of the resulting AgNPs largely depended on experimental parameters such as temperature, concentration of the Ag(I) compound, pH solution, and in the case of biological synthesis, on the direct object used to produce AgNPs [5]. As the most striking characteristic of AgNPs, their shape also largely determines their properties including the material features that these nanoparticles are part of. Despite the huge number of publications devoted to AgNP biosynthesis in bacterial, fungal, and plant cells, a more detailed approach is required not only to the synthesis itself, but also to its mechanism, the participation of various cellular compounds: proteins, enzymes, acids, etc. In addition, an important aspect to be considered in the future practical application of AgNPs are the interaction mechanism of nanoparticles directly with the cell, and the processes occurring inside it. This serious fact is extremely important, whereas the currently fashionable prefix “bio” should reflect not only the method of obtaining a practically significant substance, but also the application safety, especially used in medicine. In addition, considering silver nanoparticles as a potential medical agent, we should not forget about their potential toxicity. A large number of publications on this topic have shown that the toxic potential of nanoparticles is determined by factors such as size, shape, surface area, aggregation or agglomeration, and dose [6,7,8,9]. It is generally believed that easily ionized silver particles can affect the cell by the Trojan horse mechanism. Phagocytosis of AgNPs stimulates inflammatory signaling through the generation of reactive oxygen species (ROS) in macrophage cells, after which activated macrophage cells induce TNF-α secretion. Increased levels of TNF-α lead to cell membrane damage and apoptosis [10,11]. It should be noted that a number of studies have shown the toxicity of AgNPs (for example, in studies of rat hepatocytes and neuronal cells, mouse stem cells, and human lung epithelial cells) in relation to cells, and its absence (in studies of healthy mammalian cells) [12,13,14,15,16,17]. In this regard, the study of toxicity is extremely important using in vivo and in vitro assays as well as in silico models [18]. Therefore, this mini-review is dedicated to the mechanisms of AgNPs biosynthesis in diverse cell types (bacteria, fungi, plants) and interaction with various cells, and as a result, the multifarious use of silver nanoparticles.

## 2. Mechanism of Silver Nanoparticles (AgNPs) Biosynthesis

Notwithstanding an incredibly large number of publications describing the AgNP synthesis via various groups of organisms such as bacteria, fungi, lichens, algae, and higher plants, the mechanism of this process still remains completely unexplored.

The AgNP synthesis results using a variety of microorganisms demonstrated that the process of AgNP formation can occur both inside and outside the cell. Extracellular synthesis involves the presence of proteins–enzymes present on the cell wall of bacteria and secreted proteins, thanks to which Ag^+^ is reduced to Ag^0^. It was shown that AgNP extracellular synthesis is typical for both Gram-positive bacteria genus *Bacillus*, in particular, for *B. pumilus*, *B. persicus*, and *B. licheniformis*, *B. indicus* and *B. cecembensis* as well as *Planomicrobium *sp., *Streptomyces *sp., *Rhodococcus *sp., and for Gram-negative bacteria such as *Klebsiella pneumoniae*, *Escherichia coli*, and *Acinetobacter calcoaceticus* [19,20,21,22,23,24,25]. This mechanism of synthesis has also been determined for a number of other microorganisms such as the fungi *Rhizopus stolonifer*, *Aspergillus niger*, *Fusarium oxysporum*, *Fusarium *sp., and *A. flavus* [26,27,28,29].

Nevertheless, the nanoparticle synthesis by microorganisms has been shown in a number of studies as intracellular. This mechanism is represented in Gram-negative bacteria and is associated with membrane proteins transporting the silver ions into the cell. For example, the intracellular nature of AgNP synthesis was established for *Enterobacter cloacae* by El-Baghdady et al. [30]. Similar results were demonstrated for *Pseudomonas stutzeri* [31]. This mechanism was also shown for Gram-positive bacteria *Corynebacterium *sp. [32] as well as for various bacteria of the genus *Streptomyces* [33]. Among the representatives of the fungi kingdom, acidophilus *Verticillium* sp. can be highlighted [34]. Moreover, some microorganisms are able to perform AgNP biosynthesis both intracellularly and extracellularly including *Bacillus* strain CS 11 and *Proteus mirablis* [35,36].

Whether AgNP biosynthesis is intracellular or extracellular, the fundamental factor in this process is enzymes. Most researchers agree that it has a leading role in the AgNP formation of NADH-dependent nitrate reductase, which acts as an electron shuttle, taking electrons from the nitrate molecule and transferring it to the metal ion for the formation of nanoparticles, which is clearly shown for *F. oxysporum*, *Ps. aeruginosa*, and others [37,38,39,40,41]. The supposed mechanism of this process in shown in Figure 1. The information about respiratory [42] and periplasmic nitrate reductases [43] is also in the literature. In some experiments, it was found that proteins and sugars of the cell wall, where the bioreduction process can occur, can participate in the silver ion grab [32,44]. In addition, it is believed that the presence of a carboxylate group on the bacterial cell surface, which causes its mostly negative charge, provides an electrostatic interaction between this group and positively charged silver ions, which helps capture silver ions [45]. Some amino acids such as arginine, aspartic acid, cysteine, glutamic acid, lysine, and methionine are also implicated in the reduction of silver ions or silver nanocrystals, which act as catalysts, producing a hydroxyl ion that reacts with reducing agents such as aldehyde [46,47]. It is shown by Graf et al. that peptides containing disulfide bonds can also participate in the reduction of Ag^+^ to Ag^0^ [48]. The reaction conditions also make an important contribution: for example, a high pH plays an important role in the subunit activation of the oxidoreductase enzyme, and promotes the conversion of tryptophan into a transition tryptophil radical, which gives electrons to silver ions and leads to a reduction to elementary silver [49,50]. An important peculiarity is the intensification of AgNP biosynthesis by light. This effect may be associated with the activation of reducing agents in the culture of the supernatant, which for their part, causes the release of electrons to reduce Ag^+^ to Ag^0^ nanoparticles [51,52,53]. The other hypothetical mechanism of nanoparticle synthesis is based on the fact that certain bacteria generate the trans-membrane proton gradient, which is broken down by the active symport of Na^+^ ions along with Ag ions from the extracellular environment [54,55]. Several silver-binding membrane proteins attract silver ions and by deriving energy from ATP hydrolysis, results in the uptake of silver ions inside the cells and initiate synthesis of AgNPs [55].

An interesting fact is that the production of silver nanoparticles is possible not only with the help of nitrate reductase, but also with a completely different class enzyme: extracellular keratinase *B. safensis* plays a crucial role in AgNP biosynthesis [56,57].

Fourier-transform infrared spectroscopy (FTIR) analysis is studied for synthesized AgNPs to find out the possible reducing bio-molecules that can stabilize nanoparticles, prevent agglomeration, and create their capping in an aqueous solution. It should be noted that the final packaging of microbial nanoparticles involves a very large number of different compounds. These can be peptides, enzymes, carboxylic acids, aldehydes, ketones, rhamnose sugar, and rhamnolipids [40,58,59,60,61,62,63,64,65,66,67,68,69,70,71]. It is assumed that the enzymes can bind to silver nanoparticles using free amino and cysteine groups of proteins.

The number of publications on AgNP synthesis with the assistance of various plant extracts (leaves, stems, roots, etc.) is incalculable. A wide diversity of plants including medicinal herbs are used as “factories” for the production of silver nanoparticles. The prospective mechanism of AgNP synthesis is generally similar to that of microorganisms and is enzymatic in nature. However, the compounds for the nanoparticles’ stabilization and final capping are different and specific from those for microorganisms, because plant cells contain a complex of diversified antioxidant metabolites preventing the oxidation and damage of cellular components [72]. Therefore, enzymes, glycosides, saponins, and other biomolecules can participate in the nanoparticle’s stabilization [73,74]. They are especially important in terms of further practical applications of AgNPs, seeing that they have anti-inflammatory, antioxidant, antitumor, and other effects [75,76]. The literature data indicate that when metal salts are added to the plant extract, silver ions bind to proteins and water-soluble compounds using –OH and –COOH groups, leading to conformational changes in the protein molecule, which contribute to the captured metal ion transformation into a silver nanoparticle [77,78]. In addition, amino groups and cysteine residues of proteins take part in the silver reduction process and the formation of AgNPs [79,80]. Alkanes, amines, phenols, polyphenols, arabinose and galactose, aldehydes and ketones, alcohols, alkaloids, lignans, terpenoids, and flavonoids can act as “capping” agents for the formation of silver nanoparticles [81,82,83,84,85,86,87]. Flavonoids are particularly interesting in this case due to their high antioxidant activity for medical purposes. Hydrophilic functional groups of various compounds surrounding nanoparticles make them colloid-stable in an aqueous medium [88]. Other interesting substances that act as reducing and stabilizing substances are the sucrose and fructose of garlic extract [89]. Furthermore, polyols are responsible for the reduction of Ag^+^ into silver nanoparticles in the *Dioscorea bulbifera* tuber extract [90]. It is supposed that terpenoids are surface-active molecules that adsorb on the AgNPs surface for stabilizing nanoparticles and preventing the AgNPs from agglomeration [91]. The reduction from Ag^+^ ions to silver nanoparticles (Ag^0^) with terpenoids may involve the conversion of C–O group of the terpenes to the –C–O group [92]. It is likely that the terpenoids play a role in the reduction of metal ions by the oxidation of aldehydic groups in the molecules to carboxylic acids [93]. Apparently “capping” agents have the possibility of selective binding to different types of facets on a nanocrystal to change their specific surface free energies and thus their area proportions [94]. Thus, nanoparticle “capping” can perform several important functions, namely prevent the agglomeration of nanoparticles, reduce toxicity, and improve antimicrobial properties; additionally these molecules can enhance the affiliation possibility and action of AgNPs on the bacterial cells. [95,96]. Remarkably, plant “capping” agents frequently have their own antimicrobial activity that can increase the activity of AgNPs.

## 3. Mechanism of AgNP Action on Cells

The exact mechanism of action of silver nanoparticles on the cell is unknown. However, a significant amount of data have accumulated in this area, especially working with various plant extracts, indicating that AgNPs are able to physically interact with the cell surfaces of different bacteria. There are several bases for the AgNP effect on the cell: adhesion on the surface of the bacterial cell wall and membrane, penetration into the cell and disruption of intracellular organelles and biomolecules, induction of oxidative stress, and modulation of signal transduction pathways [97]. The adhesion and accumulation of AgNPs on the cell surface were especially observed for Gram-negative bacteria. AgNPs can penetrate bacterial cells through a water-filled channel called porins in the outer membrane of Gram-negative bacteria. Porins are primarily involved in passively transporting hydrophilic molecules of various sizes and charges across the membrane. It is likely that that the thicker cell wall of Gram-positive bacteria produce the penetration of silver ions into the cytoplasm, therefore the effect of AgNPs is more pronounced in Gram-negative bacteria than in Gram-positive bacteria [98]. It is also possible that the presence of lipopolysaccharides contributes to the structural integrity of the Gram-negative bacteria cell wall, making such bacteria more sensitive to silver nanoparticles because the negative charge of the lipopolysaccharides promotes AgNP adhesion [99]. Some researchers have assumed that the ability of silver nanoparticles to attach to the bacterial cell wall due to the electrostatic interaction between positively charged silver ions and the negatively charged surface of the cell membrane because of the carboxyl, phosphate, and amino groups, give an opportunity to subsequently penetrate it, thereby causing structural changes in the cell membrane and, as a result, its permeability. Then, dissipation of proton motive force (PMF) and thus membrane destruction occurs [100,101]. AgNPs may also act as a carrier to transport Ag^+^ more efficiently to bacteria cells whose proton motive force would consequently reduce the local pH and increase Ag^+^ release [102]. In addition, it is believed that silver nanoparticles form free radicals upon contact with bacteria that damage the cell membrane, making it porous [103].

However, other researchers are of the opinion that AgNPs adhere to the surface of bacteria and change the membrane properties, while inside the bacterial cell, they can lead to DNA damage [104,105]. For example, the review of MaQuillan et al. suggests that the primary mechanism of action of silver nanoparticles is cell membrane dissolution [106]. In addition, the dissolution of silver nanoparticles releases antimicrobial silver ions, which can interact with thiol-containing proteins in the cell wall and influence their functions. When interacting with the outer membrane, silver nanoparticles can bind to proteins, forming complexes with electronic donors containing oxygen, phosphorus, nitrogen, or sulfur atoms. It is the interaction with thiol groups that is best described in the literature. Thus, silver nanoparticles lead to the inactivation of membrane-bound enzymes and proteins by interaction with disulfide-bonds and active site blocking [107]. Reportedly, AgNPs have the possibility of increasing the trans/cis ratio of unsaturated membrane fatty acids, which leads to changes in membrane fluidity and the composition of the lipid bilayer. It can lead to changes in the membrane structure that can prevent the membrane functioning, causing an increase in permeability and loss of membrane integrity.

Moreover, as the adhesion of bacteria to any surface is the primary stage for biofilm formation, AgNPs fixed on the cell surface can prevent this process. It can be of great practical importance, especially in the fight against pathogenic microorganisms [75,108]. Additionally, the data that AgNPs contribute to the neutralization of adhesive substances involved in biofilm formation was described in [109]. Jena et al. demonstrated that the silver nanoparticles were able to mediate apoptosis of the bacteria cell by disrupting the bacterial actin cytoskeletal network [110]. The result shows that the nanoparticles affected the actin cytoskeleton MreB, causing morphological changes in the bacterial shape, thus increasing the fluidity in the membrane, which follows by rupture of the cells.

On one hand, the accumulation of AgNPs on the cell membrane results in a violation of the bilayer integrity and the appearance of breaks; on the other hand, the penetration of AgNPs directly into the cell and interaction with vital biomolecules ultimately leads to cell death. Silver ions can interact with disulfide bonds of the enzymes responsible for cellular metabolism and thiol groups, particularly respiratory enzymes, and inactivate them, generating reactive oxygen species (ROS) and cellular oxidative stress in microbes [111]. Thus, the synergistic effect of nanoparticles interacting with the cell membrane and causing the generation of ROS was revealed on *Ps. aeruginosa* and the antibacterial effect was determined [112]. ROS oxidize the double bonds of fatty acids in the membrane, which allows for the generation of other free radicals, damaging the cell membrane [113,114].

The catalytic activity of AgNPs for the formation of disulfide bonds in the oxygen molecules reaction in the cells and hydrogen atoms of the thiol groups was observed. Silver catalyzes the formation of disulfide bonds responsible for changing the shape and structure of cellular enzymes, affecting their function. Cell treatment with a 900 ppb Ag^+^ solution was found to affect the expression of some important proteins and enzymes such as the 30S ribosomal subunit, succinyl coenzyme A synthetase, maltose transporter (MalK), and fructose bisphosphate aldolase. Silver ions bind to the 30S ribosome subunit, deactivate the ribosome complex, and stop protein synthesis. The synthesis of immature precursor proteins involved in the cell membrane formation through the effect of silver nanoparticles on ribosomes, transcription, and translation, which subsequently means cell death. The AgNPs influence the important enzyme succinyl coenzyme-a-synthetase involved in the tricarboxylic acid cycle, creating cellular metabolism disturbance [115]. It was also found that the bactericidal properties of AgNPs were associated with a disruption of RNA transcription, purines, pyrimidines, and fatty acids of bacteria [116]. Suppressing various cellular metabolic processes, inhibiting nutrition, changing gene expression, affecting ATP production, and blocking the microorganism’s respiratory chain at the level of cytochrome oxidase and NADH-succinate dehydrogenase, silver nanoparticles cause oxidative stress [117]. The interaction of AgNPs with cell DNA can break its replication [118,119,120]. It was found that Ag^+^ forms complexes with nucleic acids and breaks the H-bonds between antiparallel base pairs. DNA molecule states, changing from the relaxed to condensed form, was also can caused by AgNPs, where the replication ability decreased as a result. The transcription process in microorganisms can be suppressed by intercalation of AgNPs in the DNA helix [121,122,123].

The mechanism of the relay signal necessary for microbial growth and cellular activity is represented in microbial cells by the phosphorylation cycle and the dephosphorylation cascade. AgNPs can putatively modulate cellular signaling and acts by dephosphorylating tyrosine residues on key bacterial peptide substrates, thus inhibiting microbial growth [124].

According to many studies of the silver nanoparticle synthesis in plants, AgNPs trigger the activation of the p53 protein, which acts as a suppressor of the malignant tumor’s formation in mammalian cells as well as caspase-3, which plays an important role in cellular apoptosis [125,126].

It is well-known that there exists a relationship between the size of the nanoparticles and the antibacterial effect. Smaller nanoparticles can easily penetrate the cytoplasm and the surface of the interaction of nanoparticles with both microbial cells and its components and organelles is larger [127]. The proposed mechanism of the nanoparticles’ influence is shown in Figure 2.

Thus, the importance of information about the processes of both the synthesis of nanoparticles and the mechanism of their effect on target cells for further practical application of AgNP is indisputable.

## 4. Biomedical Application of AgNPs

### 4.1. Antibacterial and Antifungal Activity

Nowadays, the resistance to many well-known antibiotics of bacteria of the genera *Streptococcus*, *Salmonella*, *Escherichia*, *Pseudomonas*, etc. is a serious medical problem that needs to be resolved as soon as possible. In the search for new bio drugs, AgNPs have been found to be a very promising area in the struggle against pathogens. The silver nanoparticles’ ability to inhibit growth and generate the death of pathogenic microorganisms that cause various types of widespread human diseases are agreed by the vast majority of research. As above-mentioned, the AgNPs’ capacity to bind with various biomolecules in microbial cells provides their persistent antibacterial effect. Today, plant extracts are an inexhaustible source of AgNP production. Often having their own therapeutic properties and forming AgNPs in a specific capping, plants are the predominant object for research in this area. Table 1 shows only a small fraction of studies that have established a comparable, and sometimes superior to antibiotics, antibacterial effect of silver nanoparticles from various plant extracts. Thus, an inhibitory effect has been shown against *Str. aureus*, *E. coli* [128], *K. pneumoniae*, *Acinetobacter baumannii* [129], *Proteus vulgaris*, *Serratia marcescens*, *Ps. aeruginosa*, *B. subtilis* [130], *Enterococcus faecalis*, *C. albicans* [131], *S. typhimurium*, *S. enteritidis* [132], *A. niger*, *A. flavus* [133], *B. cereus* [134], *Fusarium *sp., *Rhizopus sp.* [135], *F. oxysporum*, *Alternaria brassicicola* [136], *C. kefyr* [137], *Vibrio parahaemolyticus* [138], *E. aerogenes, B. bronchiseptia* [139], and *Mycobacterium tuberculosis* [140]. It is interesting that more exotic objects are used for the synthesis of AgNPs. For example, algae *Gracilaria parvispora* was used by Hussein et al. to produce silver nanoparticles that inhibited the growth of *Str. aureus* and *Ps. aeruginosa* [141], and brown algae *Sargassum longifolium* was applied for the synthesis of AgNPs against *A. fumigatus*, *C. albicans*, and *Fusarium *sp. [142]. Another peculiar source for nanoparticle production with antibacterial potential are lichens [143,144]. They are extremely interesting in this case because lichen-specific metabolites such as, for example, antranorin, may play a significant role in the synthesis of AgNPs [145].

Despite most often using plant extracts for the synthesis of silver nanoparticles, the bacteria themselves also turn out to be “biofactories” for the production of AgNPs. A bacterial “plant” for AgNPs with an inhibitory effect against pathogenic microorganisms themselves are bacteria of the genus *Bacillus* as well as other bacteria such as *E. coli*, *Brevibacterium casei*, *Str. albogriseolus*, *S. typhirium*, *Acinetobacter calcoaceticus*, *Sporosarcina koreensis*, *Aeromonas *sp., *Phenerochaete chrysosporium*, *Streptacidiphilus durhamensis*, or *Paracoccus *sp., *Ps. aeruginosa* et al. [146,147,148,149,150,151,152,153,154,155,156,157,158,159,160,161,162,163,164]. Microscopic fungi *Aspergillus*, *Penicillium*, *Fusarium*, *Trichoderma*, *C. albicans*, yeasts *Schizosaccharomyces* [165,166,167,168,169,170,171], and cyanobacteria such as *Oscillatoria limnetica, Synechococcus* sp., *Nostoc *sp., *Scytonema *sp., and *Phormidium *sp. [172,173,174] are also used for the synthesis of AgNPs. As important biotechnological objects due to their wide application for the production of various strategically important substances (antibiotics, vitamins, enzymes, biologically active substances) and due to relative simplicity in controlling the synthesis of AgNPs, microorganisms can become the main participants of green silver nanoparticle synthesis.

Currently, the combined use of AgNPs as antibacterial agents and some antibiotics is practiced in experimental medicine. However, this synergistic effect is not achieved for all antibiotics [175]. Thus, multiple drug resistant *S. typhimurium* growth was suppressed by cooperative AgNPs and well-known antibiotics enoxacin, kanamycin, neomycin, and tetracycline, while the use of ampicillin and penicillin did not give a similar result [176]. It is likely that tetracycline increases the silver nanoparticles binding to the surface of bacteria. The synergetic effect of AgNPs for other antibiotics streptomycin, vancomycin, tetracycline, amoxicillin, gentamicin, erythromycin, and ciprofloxacin against *S. aureus* and *E. coli* [177], and ampicillin, chloramphenicol, and kanamycin has also been shown against various pathogenic bacteria (*Ent. faecium, St. aureus, Str. mutans*, and *E. coli*) [178].

### 4.2. Antiviral Activity

In modern human history, viruses have been found to be one of the most terrible human disease pathogens. Despite the apparent structural simplicity, viruses reveal a huge threat in the face of dangerous diseases—Spanish influenza, HIV, Ebola, and Marburg, and finally, the 2020 pandemic caused by COVID-19—proving to us how little we know about fighting viruses. The pathogenic nature of viruses consists of attachment and penetration into the host cell. In this case, the virus binds to ligands and proteins on the cell membrane surface using its own protein components. Preventing such binding appears to be the best way of avoiding cell infection. The mechanism of AgNPs’ antiviral activity is still poorly understood, but the data available in the literature is as follows: (1) AgNPs bind to the protective coat of the protein of the virus, suppressing attachment; and (2) AgNPs bind to the virus DNA or RNA, suppressing replication or virus proliferation inside host cells [179]. For example, AgNPs have been shown to inhibit the initiation of transmitted gastroenteritis virus (TGEV) infection by binding to a surface protein, S-glycoprotein. It has been suggested that silver nanoparticles can change the structure of surface proteins, thereby reducing their recognition and adhesion to the host receptor [180]. The inhibition of the replication process in arenavirus according to silver nanoparticles was found by Speshoc et al. [181]. There is evidence of preventing host cell contact and, as a result, preventing infection for the herpes simplex virus (HSV) types 1/2, human parainfluenza virus type-3 [182], and influenza virus [183]. Moreover, AgNPs were found to be effective against human cells infected with HIV, and also effectual in adhering to the HIV envelope to prevent infection [184]. Sharma V. et al. showed a decrease in the viability of cells infected with Chikungunya arbovirus spread by two types of mosquitos: *Aedes albopictus* and *Aedes aegypti* [185]. The important research is the suppression of the COVID-19 viral activity. Thus, Sarkar D. S. suggests that silver nanoparticles can bind to virus spike glycoproteins, preventing their binding to the target cell, and also working as a weak acid that can reduce the environmental pH of the respiratory epithelium, which is the main target of the coronavirus attack, leading to its death [186]. Apparently, the size of the silver nanoparticles plays a key role in the antiviral effect (i.e., smaller particles have a more pronounced influence [187]. Further research on the AgNPs’ antiviral activity may open new possibilities in the fight against diseases invoked by various viruses.

### 4.3. Larvicidal Activity and Antiplasmodial Activity

The multiple disease distribution by mosquito vectors is one of the most serious medical problems in tropical and subtropical countries. Mosquito-borne diseases are common in more than 50 countries around the world. The most typical disease is Dengue fever, the main vector of which is the *A. aegypti* mosquito. In recent years, Dengue transmission has greatly expanded in both suburban and urban areas, becoming a global problem for millions of people. The malarial mosquitoes *Anopheles stephensi* as well as mosquitoes transferring the Zika virus, yellow fever, Japanese encephalitis, and other extremely harmful diseases are no less dangerous. The disease dissemination by mosquitoes is mainly due to ever-increasing urbanization and associated anthropogenic activities. Since effective drugs, and most importantly vaccines, have not been developed against these diseases, controlling the mosquito population in their breeding areas can be an alternative means of combating these diseases. Studies of the AgNPs’ larvicidal activity have been investigated for a long time because silver nanoparticles have many advantages such as eco-friendly drugs compared to using expensive and harmful synthetic insecticides. Most studies have demonstrated larvicidal, pupicidal, and adulticidal toxicity for *A. albopictus* and *A. aegypti* by the assistance of silver nanoparticles obtained using extracts of various plants and microbial cultures [188,189,190,191,192]. According to the hypothetical mechanism, AgNPs can penetrate the exoskeleton of young mosquitoes, and then bind to cell enzymes and DNA in the intracellular space. In addition, a membrane permeability decrease may eventually provide cell function loss and cell death [193,194]. Another trend against malaria is the AgNPs’ direct impact on the pathogen *Plasmodium falciparum* and other plasmodia [83,195,196]. Thus, larvicidal and antiplasmodial activities can make silver nanoparticles a promising factor in the fight against malaria and other tropical diseases.

### 4.4. Anthelmintic Activity

A very interesting area of AgNP application is their anthelmintic activity. Contact with the soil as well as travel to tropical regions abundant with different parasites results in human infections of various types of helminths. The most anthelmintic drugs for anthelmintic therapy act on target proteins and the activity regulation of parasite neurons and muscles, resulting in paralysis, starvation, immune attack, and expulsion of the worm. However, such drugs may have a limited activity spectrum in different types of worms and generate drug resistance [197]. The concentration-dependent nature of silver nanoparticles in such bioactivity manifestation using plant extracts has been indicated [198,199,200]. It is assumed that the lethal effect in worms is achieved by inhibiting glucose uptake and the presence of components such as glycosides, tannins, and saponins in the packaging of nanoparticles [201,202]. These phytochemicals can attach to free proteins in the gastrointestinal tract or glycoprotein on the parasite’s cuticle and cause death.

### 4.5. Leishmanicidal Activity

Leishmaniasis is another threat in equatorial countries and is a health problem. This disease is induced by 22 species of *Leishmania*, transmitted to humans by more than 90 species of sand flies [203]. Since leishmaniasis is closely linked to poverty, cramped living conditions, poor sanitation, malnutrition, and other diseases affecting the immune system, high disease rates are observed in underdeveloped and war-torn countries around the world. Although some progress in applying various strategies for the registration and treatment of this disease has been made, there is no anti-leishmaniasis vaccine, and traditional leishmaniasis treatment requires the use of toxic and poorly tolerated drugs, which are also extremely expensive and have already developed resistance. Even in this case, the use of silver nanoparticles is potentially applicable for solving the tasks set. The proposed mechanism of leishmanicidal activity is based on the generation of ROS, latency in the G0/G1 phases of the cell cycle, and inhibition of the trypanothione/trypanothione reductase enzyme system [204,205,206]. “Green” AgNPs from plant extracts with activity against *Leishmania* can be a new word in solving this serious problem.

### 4.6. Antioxidant Activity

Reactive oxygen species (ROS) such as hydroxyl, epoxyl, superoxide, peroxylnitrile, and singlet oxygen generate oxidative stress, leading to the growth of various diseases such as inflammation, atherosclerosis, aging, cancer, and neurodegenerative disorders. The antioxidant properties of a silver phyto-nanosystem make them useful in the treatment of disease. Thus, silver phyto-nanoparticles obtained from extracts of ornamental flower plants *Hyacinthus orientalis* and *Dianthus caryophyllus* (oriental hyacinth and garden clove) were found to have high antioxidant activity [207]. Salari et al. demonstrated that AgNPs synthesized using an aqueous *Prosopis farcta* fruit extract were excellent free radical “cleaners” [208]; a similar effect was shown in vitro for an aqueous extract of black Currant pomace [209], apple extract [210], leaf extracts of *Elephantopus scaber* [211], *Indigofera hirsuta* [212], and *Tinospora cordifolia* [213]. The most popular used and rapid methods for estimating antioxidant activity are the ABTS (2,2-Azino-bis (3-ethylbenzthiazoline-6-sulfonic acid radical) and DPPH (1,1-diphenyl–2–picrylhydrazyl radical) assays [214]. The high antioxidant potential of silver nanoparticles in vitro was demonstrated for the aqueous solution of a spice mixture from garlic, ginger, and cayenne pepper [215]. Authors have suggested that the high antioxidant activity of the nanoparticles may be associated with different types of functional groups from the spice mixture that were responsible for reducing and capping the AgNPs. Similar results were obtained in a number of other studies against DPPH and ABTS [141,216]. Various biologically active substances in plant extracts (polyphenols, enzymes, alkaloids, etc.) can donate hydrogen to free radicals and thus disrupt the free radical chain reaction. For example, polyphenol-capping AgNPs from the aqueous extract of *Piper longum* fruit showed antioxidant activity in vitro. Silver nanoparticles synthesized by purple sweet potato root extract (*Ipomoea batatas* L.) had radical scavenging activity in vitro. Sweet potato root extract is full of glycoalkaloids, polyphenols, and anthocyanins acting as free radical scavengers and AgNP-capping by these molecules can be great antioxidants [217]. Elemike et al. proposed that the antioxidant capacity of AgNPs was due to the presence of phenolic compounds, terpenoids, and flavonoids in plants that let nanoparticles act as singlet oxygen quenchers, hydrogen donors, and reducing agents [218]. Therefore, Shriniwas et al. supposed that the higher antioxidant activity of AgNPs from *Lantana camara* leaves may be associated with the predominant adsorption of the antioxidant substances from the extract to the surface of the nanoparticles [219]. Thus, high indicators of antioxidant phyto-nanoparticle activity may also be associated with the specific capping of AgNPs specifically for medicinal plants, whose extracts contain a variety of antioxidant substances (polyphenols, flavonoids, etc.).

### 4.7. Anti-Cancer Activity

Cancer has become a real scourge of the 20th and 21st centuries. The mass of side effects in the implemented “classical” cancer therapy and their poor tolerance became reasons for a large-scale search for new drugs of natural origin that are able to restrain disease development and cure it. Silver nanoparticles are widespread as potential agents for cancer diagnosis and therapy. There is evidence that silver nanoparticles can induce apoptosis-dependent programmed cell death in the absence of the p53 tumor suppressor. In addition, it is suggested that the cytotoxic properties of silver and hybrid silver nanoparticles may depend on the cell type as higher cytotoxicity against cancer cells was demonstrated compared to non-cancer fibroblasts [220]. Due to genetic mutations that occur in cancer, the cell cycle including a complex series of signaling pathways for the cell growth, replicates its DNA and divides, which is disrupted and leads to uncontrolled cell proliferation. Thus, the most important stages of the cell cycle such as DNA synthesis (S), Gap2/mitosis (G2/M), Gap1 (G0/G1), and subG1 are the main points for the arrest [221]. Al-Sheddi et al. showed that AgNPs produced by the plant *Nepeta deflersiana* had the ability to induce apoptosis and cell death by cell necrosis HeLA by stopping the subG1 cell cycle [222]. Silver nanoparticles were found to induce the apoptotic pathway by generating free oxygen radicals that displayed antitumor, antiproliferative, and antiangiogenic effects in vitro [223]. It was also discovered that silver nanoparticles disturb normal cellular function and influence the integrity of membranes by inducing different apoptotic signaling genes in mammalian cells, leading to programmed cell death [224]. It is well known that the high level of ROS generation can provide cellular damage by resulting in mitochondrial membrane damage, leading to toxicity [225].

Very significant achievements in the field of antitumor activity of AgNPs have been described: toxicity against tumor cells of HepG2 (human hepatocellular carcinoma) [226,227] and MCF-7 (invasive human breast ductal adenocarcinoma) [228]. It was found that the induction of apoptosis of HT29 cells (human colon cancer) can occur due to DNA fragmentation using silver nanoparticles [229]. It has also been shown that the process of apoptosis can be realized via the degradation of lysosomes during autophagy, increasing the programmed cancer cells’ death [230]. Similar results for Jurkat cells were obtained in vitro, in addition, activation of caspase-3 and condensation/fragmentation of chromatin were observed in tumor cells treated with silver nanoparticles, which led to cell death due to the apoptotic process [231]. Antitumor effect was also found for A549 cells (human adenocarcinoma) [232], HeLa cells [233], HCT116 (human colon carcinoma), MCF-7 (human breast adenocarcinoma), PC3 (prostate cell line), and A549 (lung carcinoma cell line) [234]. As in the case of antioxidant activity, the most common biofactory for the production of AgNPs is plants, especially those for which anti-cancer properties are already known. However, other organisms are also used for the synthesis of nanoparticles, for example, fungi *A. fumigatus* [234,235]. With powerful anti-carcinogenic properties and extremely low toxicity, AgNPs are extremely promising anticancer medicines.

### 4.8. Other Bioactivities

#### 4.8.1. Anti-Diabetic Activity

As alpha-amylase and a-glucosidase are key enzymes in carbohydrate metabolism, their inhibition is one of the most important strategies for diabetes therapy. Amylase and glucosidase inhibitors avoid the breakdown of carbohydrates to monosaccharides, which is the main reason for increased blood glucose levels. An amylase inhibitor, jointly with starchy foods, reduces the usual upturn in blood sugar. AgNPs are represented as alpha-amylase inhibitors in many studies in vitro and in vivo [236,237,238,239,240].

#### 4.8.2. Anti-Inflammatory Activity

In vitro, silver nanoparticles are also attributed with an anti-inflammatory effect by playing a role in the wound healing process due to TNF-α, interferons, and interleukin 1 as well as inhibition of COX-2 and MMP-3 expressions, and also have the potential to reduce the activity of TNF-α, which is involved in inflammatory processes [241,242,243,244,245]. AgNPs from the *Piper nigrum* extract were shown as selective cytokine inhibitory agents for IL-1β and IL-6 [246]. AgNPs synthesized using polyphenols present in European cranberry bush fruit extracts were developed Their anti-inflammatory effect was identified both in vitro (on HaCaT cell line, exposed to UVB radiation) and in vivo (on acute inflammation model in Wistar rats) that could be potentially used as therapeutic tools for the treatment of inflammation [247]. Silver nanoparticles from European black elderberry (Sambucus nigra) fruit extracts demonstrated an anti-inflammatory feature in vitro on HaCaT cells exposed to UVB radiation, in vivo on the acute inflammation model, and for humans on psoriasis damage. In vitro, the anti-inflammatory effects of functionalized AgNPs were indicated by the decrease in cytokine production induced by UVB irradiation, and in vivo, the pre-administration of AgNPs reduced the edema and cytokine levels in the tissues early after the induction of inflammation [248]. In addition, the synergistic effect of polyphenols and silver nanoparticles for the manifestation of anti-inflammatory activity is shown. Polyphenol-coated silver nanoparticles in vitro inhibit the production of pro-inflammatory cytokines by inhibiting activation NF-kB in macrophages, thus showing good anti-inflammatory activity in the treatment of psoriasis [249]. AgNPs produced using the *Clinacanthus nutans* water leaf extract have good analgesic and muscle relaxant properties, and can act as an analgesic agent [250]. The effect study of the AgNPs obtained from a *Cornus mas* extract in vitro on activated macrophages to simulate psoriatic skin inflammation, ex-vivo on inflamed skin biopsies obtained from psoriasis patients and, in vivo in a clinical study on patients suffering from chronic inpatient plaque psoriasis, showed that significant suppression in the production of proinflamamtory cytokines (IL-12 and TNF-α) compared to the control group during the immunohistochemistry study. It was discovered that the specific targeting of skin macrophages and suppression of the production of inflammatory mediators contributes to a substantial enhancement in the therapeutic result [251].

#### 4.8.3. Anti-Alzheimer Activity

Alzheimer’s disease is associated with AChE deficiency and AgNPs could be potential new acetylcholinesterase inhibitors. Silver nanoparticles interact with the AChE protein, inhibiting its activity, which indicates the affinity of the nanoparticles with cholinesterase. The lithophilicity of the nanoparticles and hydrophobicity environment of the enzyme ChE molecule provide this interaction [252].

#### 4.8.4. Application in Medical Equipment

AgNP coatings are used to modify catheters to prevent the formation of bacterial biofilms [253,254]. Medical dressings with silver nanoparticles were used in the clinical treatment of diverse diseases and injuries such as burns, chronic ulcers, pemphigus, and toxic epidermal necrolysis [255]. AgNPs are applied in the creation of orthopedic and orthodontic implants, dental instruments, and bandages, as well as medical clothing to avoid bacterial infections [256,257,258].

The use of silver nanoparticles for coating catheters is one of the most important areas of their medical application. Since central venous catheters (CVC) are widely used for providing access to intravenous fluids, monitoring hemodynamics, drug delivery routes, and nutritional support in critically ill patients, ensuring that they are clean and resistant to microbial contamination is a significant challenge. It was found that catheters modified with silver nanoparticles were non-toxic and capable of long-term release of bactericidal silver, which has a preventive effect against infectious complications [8,259,260]. An inhibitory effect was found against both Gram-positive (coagulase-negative staphylococci) and Gram-negative microorganisms forming a biofilm on the catheter surface coated with AgNPs [261,262].

Another important area is the application of AgNPs for the treatment of wound infections. In the last few years, wound infections caused by opportunistic microorganisms have become an important problem in modern medical practice. The main goal in overcoming the problem is rapid tissue repair processes, accompanied by maximum restoration of functionality and minimal formation of scar tissue. The wound healing process, like any complex pathophysiological mechanism includes various stages such as blood coagulation, inflammation, cell proliferation, and matrix and tissue remodeling. The accumulation of large data about the antibacterial properties of silver nanoparticles as well as the long-known bactericidal properties of silver makes it possible to use AgNPs as wound healing agents. It was found that non-toxic doses of silver nanoparticles synthesized by bacteria *Bacillus cereus* and *Escherichia fergusonii* accelerated the collagen formation and epithelization as well as slowed down angiogenesis and the length of epithelization termination in rats [263]. Data were also revealed on biomaterials for improving wound healing such as modified cotton fabrics, bacterial cellulose, and chitosan [264,265,266].

## 5. Conclusions

Thus, the low toxicity, low production cost, and multiple potential possibilities of silver bionanoparticles for solving various biological problems make them prospective objects for the following scientific research and applications in the field of practical medicine. The prospective medical applications of AgNPs obtained using a wide variety of biological objects are extremely large, and the number of publications on this topic only continues to increase from year to year. The huge variety of objects used for the synthesis of AgNPs will only expand the boundaries on silver nanoparticles. Knowledge in the biosynthesis mechanisms of these extremely interesting objects as well as the mechanisms of their influence on living organisms, will undoubtedly discover new areas of application in the near future.

## Figures and Tables

**Figure 1 jfb-11-00084-f001:**
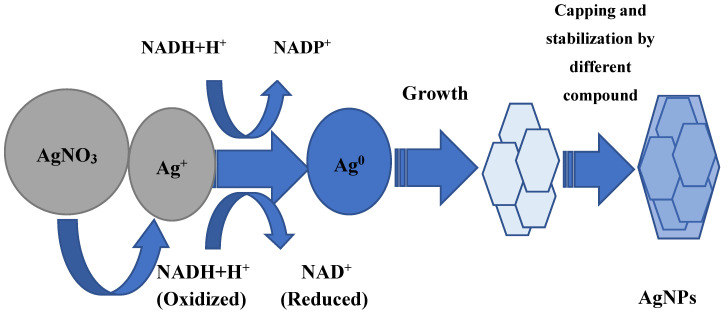
The supposed mechanism of silver nanoparticle (AgNP) biosynthesis.

**Figure 2 jfb-11-00084-f002:**
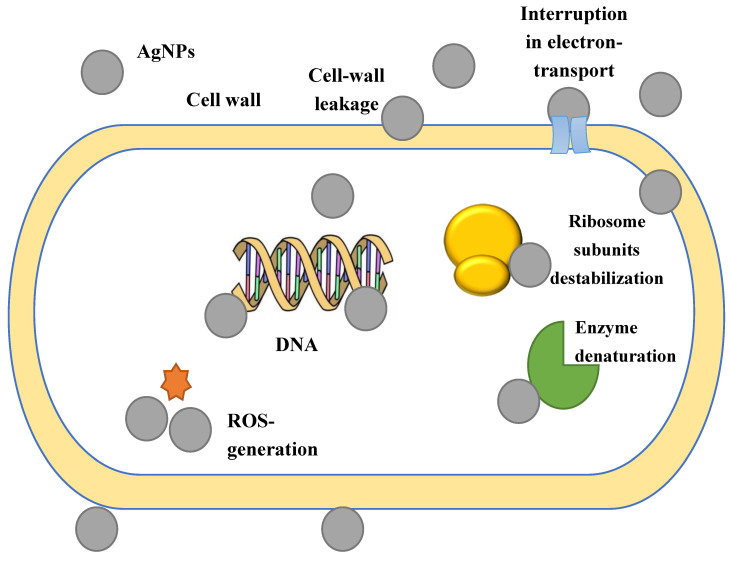
The proposed mechanism of the silver nanoparticles’ influence on bacterial cells.

**Table 1 jfb-11-00084-t001:** Antibacterial effect of silver nanoparticles from various plant extracts.

Test Microorganism	Resource of AgNPs	References
*S. aureus, Escherichia coli*	*Melissa officinalis* leaf extract	[128]
*K. pneumoniae, Acinetobacter baumannii*	*Neurada procumbens* leaf extract	[129]
*Pr. vulgaris, S. marcescens, Ps. aeruginosa, B. subtilis*	*Picea abies* L. stem bark extract	[130]
*Ent. faecalis* *and C. albicans*	*Glycyrrhiza* glabra root	[131]
*E. coli, K. pneumoniae, S. typhimurium, S. enteritidis*	pu-erh tea leaves	[132]
*St. aureus, B. subtilis, Ps. aeruginosa, E. coli, K. pneumonia, A. niger, A. flavus*	*Rhinacanthus nasutus* leaf extract	[133]
*S. aureus, B. subtilis, B. cereus, C. albicans*	Lingo-berry and cranberry juice	[134]
*Fusarium, Rhizopus, Proteus, A. flavus, A. niger*	*Svensonia hyderobadensis* leaf extract	[135]
*F. oxysporum, Alt. brassicicola*	*Citrus limon* leaf extract	[136]
*C. albicans, C. kefyr*	*Euphorbia hirta* leaf extract	[137]
*V. parahaemolyticus*	*Adathoda vasica* leaf extract	[138]
*A. fumigatus, F. solani, A. niger, A. flavus, S. aureus, E. aerogenes, B. bronchiseptia*	*Bergenia ciliate* leaf extract	[139]
*M. tuberculosis*	*Cucumis sativus* plant extract	[140]
*St. aureus, Ps. aeruginosa*	*Gracilaria parvispora* extract	[141]
*A. fumigatus, C. albicans, Fusarium sp.*	*Sargassum longifolium* extract	[142]
*St. aureus, Str. pyrogenes, Str. viridans, Corynebacterium xerosis,*	*Usnea longissima* extract	[143]
*Pr. vulgaris, Ps. aeruginosa, Ser. marcescens, S. typhi, St. epidermidis*	*Parmotrema praesorediosum* extract	[144]

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
