# Peer review of "Silver Nanoparticles: Mechanism of Action and Probable Bio-Application"

_jfb, 2020, doi:10.3390/jfb11040084_

Round 1

Reviewer 1 Report

This paper reported the information about medical application of “green” synthesized silver nanoparticles (AgNPs). The manuscript can be considered for publication after minor corrections that should be revised by the authors.

Comments:

  • In my opinion, the title of article should be slightly changed and display the fact that information about possible medical applications of “green” synthesized AgNPs was provided.
  • In lines 22-27: The same sentence is written twice.
  • In my opinion, more attention should be paid to molecules and ions which are adsorbed on surface of silver nanoparticles synthesized using “green” methods. It is important because of possible mechanisms of AgNPs action. Such molecules and ions undoubtedly affect adsorption of AgNPs on cell surface and AgNPs penetrations to bacterial cell.
  • In lines 141-143: “The silver nanoparticles ability to attach to the bacterial cell wall due to the electrostatic interaction between positively charged silver ions and the negatively charged surface of the cell membrane…” I slightly disagree with this statement. This statement displays experiments of few scientific gruops. It seems to me that if we speak about AgNPs more generally it is important to note the role of molecules or ions absorbed on surface of AgNPs.
  • A lot of information about AgNPs antibacterial properties are provided. However, it seems to me that more information about AgNPs antibacterial effect from chemical side should be added. For instance, possible effect of Ag0 atoms. I recommend few articles for the authors:

1) Durán, N., et al., Silver nanoparticles: A new view on mechanistic aspects on antimicrobial activity, Nanomedicine: Nanotechnology, Biology and Medicine, 2016. 12(3): p. 789-799.

2) Dakal, T.C., et al., Mechanistic Basis of Antimicrobial Actions of Silver Nanoparticles, Frontiers in microbiology, 2016. 7.

Author Response

Thank you very much for your attention to my article and for your positive feedback. I fully agree with your comments and tried to make all the necessary changes, which are highlighted in blue in the text (yellow indicates changes that were noted by another reviewer).

  • As for changing the title of the article, I haven't made it yet, because I don't know if this is possible at this stage of working with the article and whether it requires approval from the Editorial Board. However, I suppose it would be better to change it to "Silver nanoparticles: mechanism of action and probable bio-application". Or may be you can recommend something from your side.
  • In lines 22-27: The re-entered sentence in the text was removed.
  • Information about capping agents for silver nanoparticles that affect their antibacterial activity and other useful properties was included to the section "Mechanism of Ag-NPs biosynthesis".
  • In lines 141-143: “The silver nanoparticles ability to attach to the bacterial cell wall due to the electrostatic interaction between positively charged silver ions and the negatively charged surface of the cell membrane…” I agree that this statement cannot be taken as a dogma. This is one of the versions of the research group, but there are others. The information in the articles that you kindly provided me was extremely useful in this regard. Therefore, this information was added to the section "Mechanism of Ag-NPs action on cells".
  •  Also some data about the mechanism of antibacterial action and the chemistry of this process was entered to the section "Mechanism of Ag-NPs action on cells" because in my opinion it is most appropriate in it.

Reviewer 2 Report

Review on manuscript materials-986355 “Bio-application of silver nanoparticles”

Mikhailova was focused in this review on the biomedical application of silver nanoparticles synthetized by “green” method. This review brings data on the mechanisms involved in green synthesis and in their action on target cells. The author discussed the antibacterial, antifungal, antiviral, effects and antihelmintic and larvicidal activities of silver nanoparticles. In addition, the author discusses about the antioxidant and antitumoral effect of silver nanoparticles and review the antidiabetic and anti-inflammatory effects and the role in Alzheimer disease and in design of medical devices.  Generally, the article adheres to the journal´s standards but the authors should clarify some important issues, as follows:

  1. In Introduction section the author should add the information about the toxicity studies of silver nanoparticles. There are many articles that discuss this issue and their involvement in biomedical application.
  2. Please add new data about the anti-inflammatory role of silver nanoparticles phyto-synthetized with polyphenols from fruits in inflammation. There are several papers about experimental inflammation in rats treated with silver nanoparticles. In fact, the discussion must be comparative in vitro and in vivo.
  1. The author should insist critically on silver nanoparticles application in skin diseases (psoriasis, atopic dermatitis).
  2. The antioxidant effect is summarized and does not contain data on in vitro and in vivo effects of silver nanoparticles.
  3. The anticancer effect is only briefly mentioned and other bioactivities do no include the use of silver nanoparticles in wound healing, in the design of some catheters or in obtaining bandages. Please add this data.

Author Response

Thank you very much for your attention to my article. I fully agree with your comments and tried to make all the necessary changes, which are highlighted in yellow in the text (blue indicates changes that were noted by another reviewer).

  1. Information on the toxicity of silver nanoparticles was added in the "Introduction" section. This is indeed a very important aspect concerning the possibility of further application of silver nanoparticles.
  2. New data about the anti-inflammatory effect of silver nanoparticles in vitro and in vivo, as well as some information about the use in the treatment of psoriasis, was included.
  3. The necessary information about the antioxidant activity of AgNPs and its supposed mechanism has been added. Unfortunately, I could not find information about this type of activity in vivo for comparative analysis with in vitro experiments in the open access. 
  4. The section "Antitumor activity" has been expanded to indicate the proposed mechanism of this activity in terms of its effect on the cell cycle.
  5. Using of silver nanoparticles data for coating catheters and treating wound infections are also presented.

Round 2

Reviewer 2 Report

The authors have satisfactory answered to all questions mentioned.